# An Evaluation of CXCR4 Targeting with PAMAM Dendrimer Conjugates for Oncologic Applications

**DOI:** 10.3390/pharmaceutics14030655

**Published:** 2022-03-16

**Authors:** Wojciech G. Lesniak, Babak Behnam Azad, Samit Chatterjee, Ala Lisok, Martin G. Pomper

**Affiliations:** Russell H. Morgan Department of Radiology and Radiological Science, Johns Hopkins University, Baltimore, MD 21287, USA; bbehnam1@jhmi.edu (B.B.A.); samitchat@gmail.com (S.C.); alisok1@jhmi.edu (A.L.); mpomper@jhmi.edu (M.G.P.)

**Keywords:** CXCR4, Indium-111, PAMAM dendrimer, SPECT/CT, POL3026, biodistribution

## Abstract

The chemokine receptor 4 (CXCR4) is a promising diagnostic and therapeutic target for the management of various cancers. CXCR4 has been utilized in immunotherapy, targeted drug delivery, and endoradiotherapy. Poly(amidoamine) [PAMAM] dendrimers are well-defined polymers with unique properties that have been used in the fabrication of nanomaterials for several biomedical applications. Here, we describe the formulation and pharmacokinetics of generation-5 CXCR4-targeted PAMAM (G5-X4) dendrimers. G5-X4 demonstrated an IC_50_ of 0.95 nM to CXCR4 against CXCL12-Red in CHO-SNAP-CXCR4 cells. Single-photon computed tomography/computed tomography imaging and biodistribution studies of ^111^In-labeled G5-X4 showed enhanced uptake in subcutaneous U87 glioblastoma tumors stably expressing CXCR4 with 8.2 ± 2.1, 8.4 ± 0.5, 11.5 ± 0.9, 10.4 ± 2.6, and 8.8 ± 0.5% injected dose per gram of tissue at 1, 3, 24, 48, and 120 h after injection, respectively. Specific accumulation of [^111^In]G5-X4 in CXCR4-positive tumors was inhibited by the peptidomimetic CXCR4 inhibitor, POL3026. Our results demonstrate that while CXCR4 targeting is beneficial for tumor accumulation at early time points, differences in tumor uptake are diminished over time as passive accumulation takes place. This study further confirms the applicability of PAMAM dendrimers for imaging and therapeutic applications. It also emphasizes careful consideration of image acquisition and/or treatment times when designing dendritic nanoplatforms for tumor targeting.

## 1. Introduction

The chemokine receptor 4 (CXCR4) and its natural ligand CXCL12 play vital roles in the development of cancer, including cell survival, proliferation, and migration [1]. CXCR4 is overexpressed in various cancers, contributing to their rapid growth, metastasis, chemo-resistance, and poor prognosis [2,3,4,5]. Cancers expressing CXCR4 metastasize to tissues with high levels of CXCL12, such as bone marrow, lymph nodes, lungs, and brain, suggesting that cancer cells utilize the CXCR4/CXCL12 axis to establish metastases. Furthermore, CXCR4 is important in rheumatoid arthritis and systemic lupus erythematosus and acts as a co-receptor for HIV [6]. Accordingly, CXCR4 is an attractive diagnostic and therapeutic target [7]. Preclinical studies proved that blocking the CXCR4/CXCL12 axis with plerixafor (AMD3100), CXCR4-specific peptides, or monoclonal antibodies reduced cancer growth and metastasis while promoting sensitization to chemotherapy, radiation, or immunotherapy [2,3,7,8,9,10,11]. Clinical evaluation of CXCR4-specific, low-molecular-weight peptides and antibodies have demonstrated promising outcomes in patients with a variety of advanced malignancies [12,13,14]. To date, only a few CXCR4-targeted nanoparticles have been evaluated. Iron oxide nanoparticles conjugated with a CXCR4-specific peptide, cyclo(D-Tyr-Orn-Arg-2-Nal-Gly), provided enhanced magnetic resonance imaging contrast in CXCR4-expressing tumors [15]. AMD3100-based polymers were utilized for gene delivery and CXCR4 inhibition [16]. Polycationic, viologen-based dendrimers demonstrated inhibition of HIV1 cellular internalization via binding to CXCR4 [17]. Poly(lactic acid-co-glycolic acid) (PLGA) nanoparticles with CXCL12 on the surface were shown to inhibit the chemotaxis of THP-1 human monocytes [18]. Because they can be easily tuned, poly(amidoamine) (PAMAM) dendrimers may provide additional applications in targeted cancer treatment, such as by enabling delivery of multiple functionalities, including imaging and therapeutic agents [19,20,21,22,23,24]. PAMAM dendrimers are well-defined polymers with many reactive terminal groups, a large interior void volume, and low polydispersity. They possess the capacity for straightforward modification to achieve biocompatibility and desired properties. PAMAM dendrimers serve as versatile nanoplatforms that can be tailored to different sizes and compositions, enabling the development of nanomaterials for various biomedical applications, including targeted drug delivery, gene transfection, and imaging [19,20,21,22,23,24]. PAMAM dendrimers were also useful in targeting of CXCR4 [25,26,27]. Chittasupho et al. demonstrated that generation-4 PAMAM dendrimer conjugated with the CXCR4 antagonist, LFC131 (Tyr-Arg-Arg-Nal-Gly) and loaded with doxorubicin, selectively targeted BT-549-Luc breast cancer cells, leading to enhanced cytotoxicity and inhibition of chemotaxis, compared to control dendrimer [25]. In the following study, they demonstrated that partially acylated generation-5 PAMAM dendrimers conjugated with LFC131, bound to leukemic precursor B cells, and inhibited their migration facilitated by CXCL12 [26]. Liu et al. also reported that mixing of acylated generation 5 PAMAM dendrimers with CXCR4 agonist, a long hydrophobic peptide (GGRSFFLLRRIQGCRFRNTVDD), enhanced its solubility and bioavailability, resulting in reduced lung metastasis in a breast cancer mouse model [27]. These studies provided a strong rationale for the evaluation of in vivo tumor targeting and pharmacokinetics of CXCR4-targeted PAMAM dendrimers. We demonstrated that the polyphemusin II-derived peptidomimetic POL3026 and its PEGylated analog have suitable pharmacokinetics for non-invasive detection of CXCR4 expression in a U87-stb-CXCR4 glioblastoma mouse model [28]. Here, we report on the application of POL3026 to generate CXCR4-targeted PAMAM dendrimers (G5-X4) and their evaluation, along with control dendrimers (G5-Ctrl), in mice bearing U87-stb-CXCR4 and U87 tumors with high and low CXCR4 expression, respectively. Both CXCR4-targeted and control dendrimers showed similar long circulation times and a degree of passive tumor accumulation. Accumulation was further enhanced for G5-X4 through active targeting in U87-stb-CXCR4 tumors as demonstrated by biodistribution and single-photon emission computed tomography/computed tomography (SPECT/CT) studies.

## 2. Materials and Methods

### 2.1. Materials

All chemicals were purchased from Sigma-Aldrich (St. Louis, MO, USA) or Fisher Scientific (Hampton, NH, USA) unless otherwise specified. Amine terminated ethylenediamine core generation 5 poly(amidoamine) dendrimer (G5(NH_2_)_128_) was obtained from Dendritech, Inc. (Midland, MI, USA). POL3026 peptide was synthesized by CPC Scientific (Sunnyvale, CA, USA) with >95% purity. Piperidine, diisopropylethylamine (DIPEA), *N*-(3-Dimethylaminopropyl)-*N*′-ethylcarbodiimide hydrochloride (EDC), and 1-hydroxybenzotriazole (HOBt) were obtained from ChemImpex International Inc. (Wood Dale, IL, USA). ^111^InCl_3_ (t_1/2_ = 2.81 days) and DOTA-NHS-ester were acquired from Nordion (Ottawa, ON, Canada) and Macrocyclics (Plano, TX, USA), respectively. Fmoc-N-PEG_2000_-NHS-ester and mPEG_2_-NHS-ester were purchased from Creative Pegworks (Chapel Hill, NC, USA) and Quanta Biodesign (Plain City, OH, USA), respectively. All reagents and solvents were used as received without further purification.

### 2.2. Modification of POL3026 with the Linker

First, 0.030 g of POL3026 (Figure 1A, 10.7 µmole) was dissolved in 1 mL DMF, and 0.1 mL DMF containing 1.3 mg succinic anhydride (SA, 10.3 µmole) was added. The pH was adjusted to 7.0 using diisopropylethylamine (DIPEA), and the reaction was carried out for 3 h at room temperature (RT). Obtained SAPOL3026 was purified on reverse-phase high-performance liquid chromatography (RP-HPLC) as described earlier [28]. SAPOL3026 eluted at ~41.5 min was collected, evaporated, dissolved in deionized water, and lyophilized, yielding 0.026 g of the product as a white powder, which was used for further experiments. The resulting SAPOL3026 was analyzed by electrospray ionization mass spectrometry. Theoretical chemical formula: C_99_H_144_N_32_O_23_S_2_, exact mass: 2213.05, molecular weight: 2214.53, observed *m*/*z*: 2212.78—(M + 1)^+1^, 1107.13—(M + 2)^+2^/2, 738.44—(M + 3)^+3^/3.

### 2.3. Synthesis of CXCR4-Targeted G5-X4 and Control G5-Ctrl Dendrimers

Preparation of the targeted and control dendrimers involved a multi-step synthesis as presented in Figure 1B: 0.205 g of G5(NH_2_)_128_ dendrimer (MW = 28,826 Da, 6.94 µmole) was dissolved in 10 mL DMSO, and 1 mL DMSO containing 4 mole equivalent of DOTA-NHS-ester (0.021 g, 27.7 µmole) was added with vigorously stirring. The resulting reaction mixture was stirred at RT for 4 h, and the solvent was removed on a rotary evaporator. The obtained residue was dissolved in 15 mL of PBS and transferred to an Amicon centrifugal filter with 10 kDa molecular weight cut-off (MWCO), which was centrifuged for 30 min at 4000× *g* and RT. This process was repeated twice with PBS buffer and 6 times of deionized water. The purified product was dissolved in deionized H_2_O and lyophilized that provided 0.218 g of G5(NH_2_)_124_(DOTA)_4_ dendrimer (**2**). In the next step, 0.18 g of **2** (6.51 µmole) was reacted with 10 mole equivalent of Fmoc-NH-PEG_2000_-NHS-ester (0.13 g) in 10 mL of DMSO for 4 h, followed by evaporation of the solvent on a rotary evaporator and purification as described above. Then, 0.281 g of G5(NH_2_)_116_(DOTA)_4_(PEG_2000_-NFmoc)_4_ conjugate (**3**) was obtained. Then, 0.23 g of the conjugate **3** (5.22 µmole) was reacted with 2 mole equivalent (on the basis of remaining terminal -NH_2_ groups) of methyl ether polyethylene glycol 2-NHS ester (mPEG_2_-NHS, 0.148 g) in 10 mL DMSO. After mixing, the pH of the reaction mixture was adjusted to 10 with DIPEA. After stirring for 16 h at RT, DMSO was evaporated, and the G5(NH_2_)_52_(DOTA)_4_(PEG_2000_-NFmoc)_4_(mPEG_2_)_64_ conjugate was purified using ultrafiltration as described above. Then, the Fmoc protecting group was removed from PEG_2000_ using 50% piperidine/DMF solution and purified using Amicon centrifugal filters as described above to obtain non-targeted control G5-Ctrl dendrimer conjugated with an average of 4 DOTA molecules, 62 mPEG_2_ capping agents, and 8 PEG_2000_-NH_2_ moieties (based on the main generation), as measured by matrix-assisted laser desorption/ionization time-of-flight mass spectrometry (MALDI-TOF MS). G5-Ctrl was further conjugated SAPOL3026 to generate a CXCR4-targeted G5-X4 dendrimer. Then, 0.011 g of SAPOL3026 (3.96 µmole) was dissolved in 2 mL DMF followed by the addition of 2 mole equivalent of EDC and HOBt to activate the carboxyl group. After 30 min, DMSO containing 0.02 g of G5-Ctrl (0.39 µmole) was added, and pH was adjusted to 7.4 using DIPEA. The resulting reaction mixture was stirred at room temperature for 16 h, followed by evaporation of the solvent on a rotary evaporator and purification of G5-X4 as described above. Starting G5(NH_2_)_128_ dendrimer, intermediated products, and final products were characterized by MALDI-TOF MS, reverse phase high performance liquid chromatography (RP-HPLC), ^1^H NMR, dynamic light scattering, and zeta potential analysis (Appendix A). 

### 2.4. Matrix-Assisted Laser Desorption Ionization-Time-of-Flight

MALDI-TOF spectra were acquired on a Voyager DE-STR spectrophotometer. Matrix (2,5-dihydroxybenzoic acid), starting dendrimers, and synthesized conjugates were dissolved in a 50% MeOH aqueous solution containing 0.1% TFA. Equal volumes of the matrix (10 µL, 20 mg/mL) and dendrimers (4 mg/mL) were mixed, and 1 µL of the resulting mixture was spotted on the target plate, evaporated, and used for data collection. 

### 2.5. Dynamic Light Scattering (DLS) and Zeta Potential (ZP) 

DLS and zp analyses were carried out using a Malvern Zetasizer Nano ZEN3600. G5(NH_2_)_128_ dendrimer and all synthesized conjugates were prepared in 1xPBS at a concentration of 2 mg/mL Results in Table 1 are a mean of three sequential measurements of number weighted size distribution, as it most accurately represents the size of nanoparticles present in the solution and their content. 

### 2.6. Cell Lines

All cell culture reagents were obtained from Invitrogen (Halethorpe, MD, USA) unless otherwise indicated. The CHO (Chinese hamster ovary), U87 (human glioblastoma), H1155 (non-small cell lung cancer (NSCLC), and H69 (human small cell lung cancer SCLC) cell lines were purchased from American Type Culture Collection (Manassas, VA, USA). A human U87 cell line stably transfected with human CXCR4 (U87-stb-CXCR4) and CD4 was purchased from the NIH AIDS Research Reference Reagent Program [29]. All cell lines were cultured per manufacturer protocols. The CHO cell line with stable expression of SNAP-CXCR4 (CHO-SNAP-CXCR4) was generated in our laboratory and maintained as described earlier [30].

### 2.7. Competitive Binding Assays

The ability of G5-X4 and G5-Ctrl dendrimers to inhibit fluorescently labeled CXCL12-Red binding to CXCR4 was evaluated using frequency resonance energy transfer (FRET) assay as described previously [30]. Concentrations of G5-Ctrl and G5-X4 ranged from 0.1 pM to 10 µM. IC_50_ values were derived by fitting the data to sigmoidal dose–response curve using Prism 8 Software (GraphPad) and converted to *K*_i_ using the Cheng–Prusoff equation with *K*_D_ = 19 for CXCL12-Red and its concentration of 15 nM. 

### 2.8. Flow Cytometry 

H69 small cell lung cancer (SCLC) cells were detached from the growing flask using a non-enzymatic cocktail (Gibco) when confluency reached 50–70%. Detached cells were washed twice with 5 mL of 1xPBS buffer containing 2 mmol EDTA and 0.5% FBS, which was further used for flow cytometry. CXCR4 expression was evaluated using the allophycoerythrin -conjugated anti-human CXCR4 monoclonal antibody (clone12G5, R&D Systems) per manufacturer instructions. After immunostaining of CXCR4, H69 cells were examined on a FACSCalibur flow cytometer (Becton Dickinson, NJ, USA). Obtained flow cytometry results were analyzed using FlowJo software.

### 2.9. Inhibition of Chemotaxis

The effect of G5-X4, G5-Ctrl, and POL3026 on CXCR4/CXCL12-mediated chemotaxis of H69 small cell lung cancer (SCLC) cells with high CXCR4 expression [31] was evaluated using a CytoSelect cell migration assay (Cell Biolabs, Inc. San Diego, CA, USA) according to the manufacturer’s protocol. Briefly, 50,000 cells were suspended in 100 µL RPMI containing G5-X4, G5-Ctrl and POL3026 at concentration of 10 or 100 nM and transferred into insert wells. The insert wells were placed inside the harvesting wells containing RPMI with 100 nM CXCL12 (Peprotech, Rocky Hill, NJ, USA, product # 300-28A). After 17 h incubation at 37 °C, cells that migrated in the harvesting wells were lysed, CyQuant GR dye was added and the fluorescence was measured using Perkin Elmer-2480 Automatic Gamma Counter (PerkinElmer, Waltham, MA, USA).

### 2.10. Radiolabeling 

The radiolabeling of G5-Ctrl and G5-X4 with ^111^In (t_1/2_ = 2.81 days) was carried out in 0.1 M sodium acetate (pH 4.5) for 1 h at 37 °C. To remove loosely bound and unchelated ^111^In^3+^, ethylenediaminetetraacetic acid (EDTA) was added to the reaction mixture to achieve a concentration of 5 mM, followed by an additional 5 min incubation at RT and purification on a Zeba™ spin desalting column with 7 kDa MWCO (0.5 mL), then pre-equilibrated with PBS. The radiochemical purity of the resulting [^111^In]G5-Ctrl and [^111^In]G5-X4 was 98% as determined using instant thin-layer chromatography (ITLC) with citrate–phosphate–dextrose solution as the mobile phase. 

### 2.11. Animal Models

All experimental procedures using animals were conducted according to protocols approved by the Johns Hopkins Animal Care and Use Committee. A total of 35 female non-obese diabetic/severe combined immunodeficiency NOD/SCID or 5 NSG mice, from six to eight weeks old, were purchased from The Johns Hopkins Immune Compromised Animal Core. NOD/SCID mice were implanted subcutaneously (s.c.) with U87 and U87-stb-CXCR4 cells (4 × 10^6^ cells/100 μL) in the opposite flanks. Animals were used for ex vivo biodistribution and SPECT/CT imaging studies when the tumor size reached approximately 200–400 mm^3^. An orthotopic mouse model of non-small cell lung cancer was generated by injecting H1155 cells into the left lung of NSG mice using the following surgical procedure. Mice were anesthetized with 2% vaporized isoflurane. An approximately 1 cm incision was made on the skin near the left scapula, and the costal layer was exposed by separating the thoracic muscles. One million H1155 cells (in 30 µL HBSS with 50% matrigel) were injected into the left lung through the intercostal space, using a 29 G needle. Skin incisions were sutured, antibiotics were applied topically, and mice were kept on a heat-pad under observation until fully recovered from anesthesia. Tumor growth was monitored with computed tomography (CT) imaging recorded using an X-SPECT small animal SPECT/CT system (Gamma Medica Ideas, Northridge, CA, USA) and 512 projections.

### 2.12. Ex Vivo Biodistribution 

NOD/SCID mice bearing U87 and U87-stb-CXCR4 xenografts were injected intravenously with 0.74 MBq (20 μCi) of [^111^In]G5-X4 or [^111^In]G5-Ctrl formulated in 200 μL of sterile PBS. Mice were sacrificed at 1, 3, 24, 48, and 120 h post injection, and blood, tumors, and selected organs were dissected and weighed. The radioactivity in the collected samples was measured using a Perkin Elmer-2480 Automatic Gamma Counter (PerkinElmer, Waltham, MA, USA). In the blocking experiment, mice were injected with 2.33 mg/kg (45 μg) of unmodified POL3026, 60 min prior to intravenous injection of [^111^In]G5-X4 or [^111^In]G5-Ctrl. To calculate the percentage of injected dose per gram of tissue (%ID/g), triplicate radioactive standards (10% of the injected dose) were counted along with samples. 

### 2.13. SPECT/CT Imaging and Analysis

For whole-body SPECT/CT imaging, NOD/SCID mice bearing U87-stb-CXCR4 and U87 tumors were intravenously injected with approximately 12.95 MBq (350 uCi) of [^111^In]G5-X4, and images were acquired on an X-SPECT small animal SPECT/CT system (Gamma Medica Ideas, Northridge, CA, USA) as described previously [32]. Prior imaging mice were anesthetized with 2% and maintained under 1.5% of isoflurane (*v*/*v*) during the acquisition of images. Images were acquired 2 and 8 h after radiotracer injection using 64 projections over 360° at 45 s per projection and medium energy pinhole collimators. CT was recorded in 512 projections to allow anatomic co-registration. Data were reconstructed using the ordered subsets-expectation maximization algorithm, and 3D-volume-rendered images were generated using Amira 5.3.0 software (Visage Imaging Inc. San Diego, CA, USA). The same procedure was used to monitor the accumulation of [^111^In]G5-Ctrl in mouse bearing orthotopic H1155 NSCLC tumor.

### 2.14. Data Analysis

Statistical analysis was performed using a Graphpad Prism 8 software and an unpaired two-tailed t-test. When *p* < 0.05, the difference between the compared groups was considered statistically significant. 

## 3. Results

### 3.1. Synthesis and Physicochemical Characterization of G5-Ctrl and G5-X4

Syntheses of the CXCR4-targeted (G5-X4) and control (G5-Ctrl) dendrimers are presented in Figure 1. Amine-terminated generation-5 PAMAM dendrimer G5-(NH_2_)_128_ was consecutively conjugated with, on average based on main generation dendrimers, four DOTA chelators (**1**) to enable radiolabeling, eight PEG_2000_-NH-Fmoc linkers (**2**) for attachment of targeting moieties and 62 m-PEG_2_ short capping agents (**3**) to reduce net surface positive charge, followed by deprotection of PEG_2000_-NH-Fmoc (**4**), which generated the G5-Ctrl dendrimers. G5-Ctrl was subsequently conjugated with three SA-POL3026 peptidomimetics (**5**) to formulate the CXCR4-targeted G5-X4 dendrimers. To enable covalent attachment of POL3026 with G5-Ctrl, the lysine side chain of the peptide was selectively carboxylated with succinic anhydride. Resulting SA-POL3026 was conjugated to a PEG_2000_-NH_2_ linker via carbodiimide coupling. Starting dendrimer and synthesized conjugates were characterized by MALDI-TOF, RP-HPLC, ^1^HNMR, DLS, and zeta potential (Table 1 and Appendix A). Commercially available PAMAM dendrimers contain three different species trailing generation, main generation, and dimers [33,34]. All three species were detected in the spectrum of the dendrimers that we used as starting material (Appendix A), with the most pronounced peak for main generation-5 dendrimer at 26,111 Da. Trailing generation and dimers were detected at 12,600 and 50,950 Da, respectively. An increase in the molecular weights indicated conjugation of on average two, four, and five DOTA molecules with trailing generation, main generation, and dimers, respectively. Starting from conjugates III trailing generation, main generation-5 and dimers could not be resolved, showing a broad peak indicating a wide distribution of conjugates present in the sample. A shift of the highest signal intensity detected for this peak was used to calculate the average number of covalently attached PEG_2000_, mPEG_2_, and SA-POL3026 moieties. In agreement with MALDI-TOF, RP-HPLC showed the presence of trailing generation, main generation, and dimers with relative percent of 3.5, 90.9, and 5.6%, respectively, in dendrimers used as starting material (Appendix A). RP-HPLC of G5-Ctrl and G5-X4 showed a significant broadening of the peaks and shift to longer retention with no separation of trailing generation, main generation, and dimers, confirming broad distribution of the conjugates. ^1^H NMR further confirmed modifications of dendrimers (Appendix A), DLS analysis indicated an increase in dendrimer size upon PEGylation by approximately 3 nm and a significant reduction in the net surface positive charge (Appendix A). G5-Ctrl and G5-X4 dendrimers had size distribution with 7.81 ± 0.85 nm and zeta potential of 12.41 ± 1.43 mV and 9.11 ± 0.92 nm and zeta potential of 14.93 ± 0.51 mV, respectively. 

### 3.2. Evaluation of In Vitro CXCR4 Specificity

To test the CXCR4 binding affinity of G5-X4 and G5-Ctrl, competitive binding assays were carried out against CXCL12-Red (Figure 2A). The binding isotherm obtained for G5-X4 showed a concentration-dependent displacement of CXCL12-Red from CXCR4 with a half maximal inhibitory concentration (IC_50_) of 0.95 nM, (95% confidence intervals 0.6 nM–0.15 nM) and inhibition constant (*K*_i_) of 0.33 nM (95% confidence intervals 0.31 nM–0.82 nM), indicating high affinity. In contrast, the G5-Ctrl did not displace CXCL12-Red in concentrations ranging from 0.1 pM to 10 µM. We also evaluated the inhibition of CXCR4/CXCL12-mediated chemotaxis of H69 SCLC cells with high CXCR4 expression by POL3026, G5-X4, and G5-Ctrl (Appendix A) [31]. G5-X4 was superior with 93.72 ± 3.45% and 92.46 ± 3.53% inhibition of H69 SCLC cells migration at 100 and 10 nM, respectively, compared to 48.81 ± 8.18% and 31.52 ± 14.1% obtained for POL3026 and 36.61 ± 6.51% and 26.16 ± 7.75% obtained for G5-Ctrl at the same concentrations. 

### 3.3. Radiolabeling of G5-Ctrl and G5-X4

To evaluate the pharmacokinetics of the G5-Ctrl and G5-X4 dendrimers, they were radiolabeled with ^111^In. Resulting [^111^In]G5-Ctrl and [^111^In]G5-X4 radiotracers were prepared with a specific activity of 2 mCi/mg ± 0.1 mCi and radiochemical purity > 95% as confirmed by instant thin-layer chromatography.

### 3.4. Evaluation of In Vitro Specificity

To evaluate the CXCR4 specificity of [^111^In]G5-X4, we performed in vitro binding assays using U87-stb-CXCR4 and U87 cell lines with high and low expression of CXCR4, respectively (Figure 2B), as described earlier [35]. [^111^In]G5-X4 demonstrated higher uptake in U87-stb-CXCR4 cells (16.92 ± 4.83%ID, incubated dose) compared to U87 cells (3.38 ± 0.08%ID). The specific binding of [^111^In]G5-X4 to U87-stb-CXCR4 cells could be blocked with 1 µM of unmodified POL3026 peptidomimetic. [^111^In]G5-Ctrl showed relatively lower uptake 3.28 ± 0.23%ID and 4.32 ± 0.52%ID in U87 and U87-stb-CXCR4 cell lines, respectively, that did not change in the presence of POL3026. 

### 3.5. Biodistribution of [^111^In]G5-X4 and [^111^In]G5-Ctrl

Biodistribution of [^111^In]G5-X4 and [^111^In]G5-Ctrl dendrimers was carried out in mice bearing U87 and U87-stb-CXCR4 tumors with low and high CXCR4 expression, respectively (Figure 3A,B). [^111^In]G5-X4 showed accumulation in U87-stb-CXCR4 tumors with 8.2 ± 2.1, 8.43 ± 0.52, 11.52 ± 0.91, 10.44 ± 2.6, and 8.82 ± 0.51%ID/g at 1, 3, 24, 48, and 120 h after injection, respectively (Figure 3A). In U87 tumors, these values were 4.82 ± 1.51, 4.37 ± 0.31, 9.69 ± 1.2, 8.01 ± 1.53, and 6.72 ± 0.32%ID/g at the same time points. [^111^In]G5-Ctrl showed similar uptake in 87-stb-CXCR4 and U87 tumors averaging at 4.56 ± 0.67, 5.02 ± 0.88, 8.44 ± 1.67, 10.06 ± 1.46, and 7.10 ± 0.15 ID%/g at 1, 3, 24, 48, and 120 h after injection, respectively (Figure 3B). U87-stb-CXCR4/U87 (targeted-tumor/control tumor) ratios were higher for [^111^In]G5-X4 compared to [^111^In]G5-Ctrl in particular at 1 h 1.83 ± 0.13 vs. 1.25 ± 0.22 and 3 h 1.93 ± 0.02 vs. 1.35 ± 0.11, respectively (Figure 3C), suggesting relatively faster uptake of [^111^In]G5-X4 in CXCR4-expressing tumors. At 24 h, these ratios were similar for both radiotracers and increased at 48 and 120 h for [^111^In]G5-X4, indicating its longer retention in CXCR4 positive tumors. U87-stb-CXCR4/muscle ratios were high for both [^111^In]G5-X4 and [^111^In]G5-Ctrl radiotracers as the radioactivity accumulation in muscles was below 2% at all time-points evaluated. CXCR4-targeted and control dendrimers showed similar long circulation times with 44.78 ± 7.19 ID%/g at 1 h, 30.51 ± 3.21 ID%/g at 3 h, 21.41 ± 1.39 ID%/g at 24 h, 7.73 ± 1.29 ID%/g at 48 h, and 3.29 ± 1.12 ID%/g at 120 h after injection for [^111^In]G5-X4 and 38.24 ± 2.38, 26.38 ± 3.37, 13.05 ± 4.23, 10.56 ± 2.54, and 2.38 ± 0.19 ID%/g for [^111^In]G5-Ctrl at the same time points. The biodistribution of targeted and control dendrimers in off-target tissues was similar, with the highest accumulation in the liver reaching approximately 13 %ID/g at 24, 48, and 120 h, followed by spleen, kidneys, and bladder. Relatively high radioactivity was also detected in lungs and heart at early time points, which gradually decreased as the concentration of [^111^In]G5-X4 and [^111^In]G5-Ctrl in the circulation declined by over 120 h. In agreement with biodistribution, SPECT/CT imaging showed higher uptake of [^111^In]G5-X4 in U87-stb-CXCR4 tumors compared to U87 tumors and high background (Figure 3D). In mice that received a blocking dose of POL3026 1 h prior to administration of radiolabeled dendrimers, accumulation of [^111^In]G5-X4 in U87-stb-CXCR4 tumors decreased from 8.2 ±2.1 to 5.05 ± 0.32%ID/g (Figure 4A,B, *p* < 0.01), which was similar to radioactivity levels detected in U87 tumors and to [^111^In]G5-Ctrl accumulation in U87 and U87-stb-CXCR4 tumors. The blocking dose of POL3026 did not affect the blood pool concentration and overall biodistribution of [^111^In]G5-X4 and [^111^In]G5-Ctrl. We also evaluated the accumulation of [^111^In]G5-Ctrl in mice bearing orthotopic H1155 NSCLC tumors (Appendix A). SPECT/CT images indicated higher [^111^In]G5-Ctrl uptake in the tumor compared to lungs over 120 h. Ex vivo biodistribution confirmed a relatively high accumulation of [^111^In]G5-Ctrl in the tumor with 21.49%ID/g. 

## 4. Discussion

PAMAM dendrimers can be readily tailored to different sizes and compositions depending on the application. Here, we focused on the formulation and evaluation of CXCR4-targeted PAMAM dendrimers. Pharmacokinetics, biodistribution, and tumor targeting of PAMAM dendrimers are strongly influenced by their size and surface properties [36,37,38]. PAMAM dendrimers smaller than 5 nm (up to generation-4) form flexible scaffolds that are rapidly excreted by renal filtration [36,37,38]. Starting from generation 5 (size ≥ 5 nm), dendrimers have more rigid and globular structures and exhibit longer circulation times with hepatobiliary excretion in addition to renal clearance [39]. To synthesize control and CXCR4-targeted dendrimers, we have utilized generation-5 PAMAM dendrimers. To extend circulation for enhanced tumor accumulation and to reduce uptake by the reticuloendothelial system (RES), dendrimers were extensively PEGylated with PEG_2000_ and mPEG_2_, which increased their size by approximately 3 nm. As a targeting moiety, we have used the POL3026 peptidomimetic, as it was initially reported by our group to have suitable pharmacokinetics for in vivo targeting and imaging of CXCR4 [28]. G5-X4 dendrimers showed low nanomolar in vitro CXCR4 affinity, with an IC_50_ value comparable to previously reported for PEGylated POL3026 [28]. G5-X5 also demonstrated superior inhibition of H69 SCLC cell migration toward CXCL12 compared to PLO3026 and G5-Ctrl, confirming the potential therapeutic applications of this construct. In vitro CXCR4 specificity of G5-X4 was further reflected by higher uptake of [^111^In]G5-X4 in U87-stb-CXCR4 cells compared to U87 cells, which was blocked by unmodified POL3026. In contrast, G5-Ctrl nanoparticles neither inhibited the interaction between CXCR4 and CXCL12 nor showed preferential uptake by U87-stb-CXCR4 cells. Our biodistribution results showed that accumulation of [^111^In]G5-X4 in U78-stb-CXCR4 tumors was nearly 50% higher compared to U87 tumors and then [^111^In]G5-Ctrl in both tumors at 1 h and 3 h after injection, suggesting faster uptake of targeted dendrimers facilitated by CXCR4. These findings were also supported by SPECT/CT imaging, showing a higher uptake of radioactivity in U87-stb-CXCR4 tumors vs. U87 tumors at early time-points in mice treated with [^111^In]G5-X4. This initially increased uptake of [^111^In]G5-X4 in U87-stb-CXCR4 tumors could be blocked by POL3026, further confirming interactions of targeted dendrimers with CXCR4. [^111^In]G5-X4 showed similar uptake in CXCR4 positive and negative tumors at comparable levels detected for [^111^In]G5-Ctrl at 24 h that is most likely attributed to prolonged circulation and significant passive tumor targeting due to the enhanced permeability and retention (EPR) effect [40]. [^111^In]G5-X4 showed higher accumulation in U87-stb-CXCR4 tumors compared to U87 tumors and [^111^In]G5-Ctrl at 48 and 120 h after injection, suggesting that interactions of targeted dendrimers with CXCR4 could prolong their tumor retention. Similar trends with significant passive and minor active tumor targeting were demonstrated for prostate-specific membrane antigen (PSMA)-targeted polylactic acid (PLA) and non-targeted control nanoparticles with a size of approximately 100 nm [41]. In contrast, generation-4 PAMAM dendrimer-conjugated lysine-glutamate-urea derivative for PSMA targeting with a size of approximately 5 nm showed mostly active targeting, reflected by significantly higher uptake in PSMA^+^ PC3 PIP tumors compared to PSMA^-^ PC3 flu tumors [35]. [^111^In]G5-X4 and [^111^In]G5-Ctrl also showed persistent accumulation in the liver and spleen, indicating their uptake by the RES system despite extensive PEGylation. In the lungs, heart, and kidneys, initial high concentration of [^111^In]G5-X4 and [^111^In]G5-Ctrl gradually decreased as the blood pool radioactivity declined. We have recently demonstrated that non-targeted generation-4 hydroxy terminated PAMAM dendrimers with size around 5 nm below renal filtration cut-off quickly cleared from bloodstream with major accumulation in kidneys, followed by liver and very low (below 1% ID/g) uptake in PSMA^+^ PC3 PIP and PSMA^−^ PC3 flu prostate tumor models [35]. In contrast, our [^111^In]G5-Ctrl control dendrimers with size averaging around 8 nm showed significantly longer circulation and substantial passive tumor accumulation, reaching 10.06 ± 1.46%ID/g at 48 h after injection in U87-stb-CXCR4 and U87 tumors. Biodistribution of [^111^In]G5-Ctrl is similar to the previous report on hydroxy-terminated generation-7 PAMAM dendrimers with size around 7 nm, which showed approximately 7 %ID/g accumulation in orthotopic A2780 ovarian tumors due to passive targeting [39]. Taken together, our results indicate that an increase in dendrimer size by only 3 nm has a profound effect on their pharmacokinetics and passive tumor accumulation. The tumor accumulation of non-targeted dendrimers with a size of approximately 8 nm ranges between 7 and 10 %ID/g, which is comparable to the tumor uptake of PEGylated liposomes or PLA micelles with sizes of 120 nm and 100 nm, respectively [41,42]. Our results also demonstrated relatively high uptake of [^111^In]G5-Ctrl in the orthotopic H1155 NSCLC mouse model, further supporting its potential in passive tumor targeting for imaging and therapeutic applications. 

## 5. Conclusions

Our results demonstrate nearly a two-fold enhancement of targeted dendrimers in CXCR4-positive tumors, at earlier time points, compared to control tumors. That uptake differential decreased substantially 24 h after injection. The pharmacokinetics and CXCR4-targeted active tumor uptake of PAMAM dendrimers can be improved through optimization of size and surface properties. In the future, such constructs may be outfitted with imaging agents, drugs, or nucleic acids to detect and treat cancer.

## Figures and Tables

**Figure 1 pharmaceutics-14-00655-f001:**
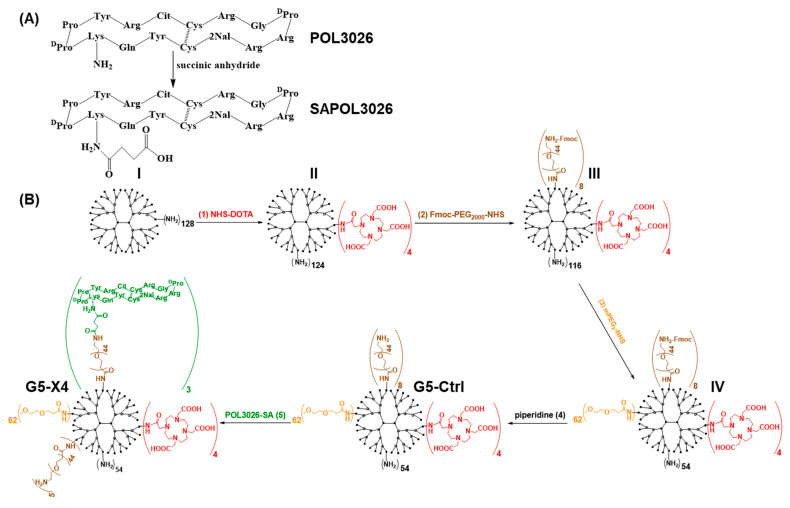
Synthesis of control G5-Ctrl and CXCR4-targeted G4-X4 dendrimers. (**A**) Selective modification of POL3026 with succinic anhydride. (**B**) Synthetic pathway leading to formation of control G5-Ctrl and CXCR4-targeted G5-X4 dendrimers.

**Figure 2 pharmaceutics-14-00655-f002:**
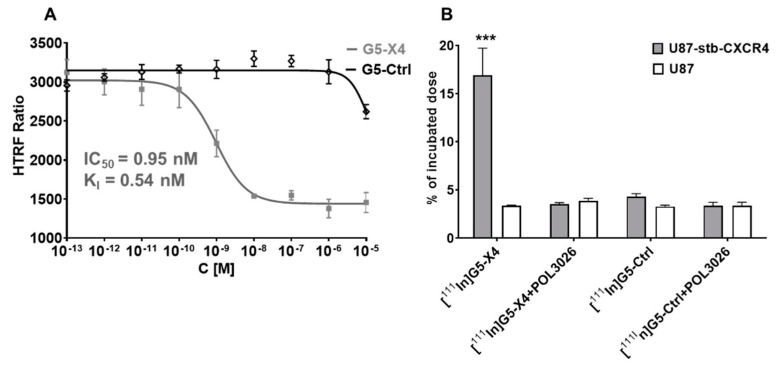
In vitro evaluation G5-X4 and G5-Ctrl. (**A**) Assessment of CXCR4 affinity: CHO-K1-stb-CXCR4-Lumi4-Tb cells were incubated with CXCL12-Red and increasing concentration of either G5-X4 or G5-Ctrl at RT for 2 h followed by FRET analysis. (**B**) In vitro binding: U87-stb-CXCR4 or U87 cells were incubated with [^111^In]G5-X4 or [^111^In]G5-Ctrl in the presence or absence of POL3026 for 1 h at RT, washed and analyzed on a gamma counter. Both assays indicate high in vitro CXCR4 specificity of G5-X4 dendrimer. *** *p* < 0.01.

**Figure 3 pharmaceutics-14-00655-f003:**
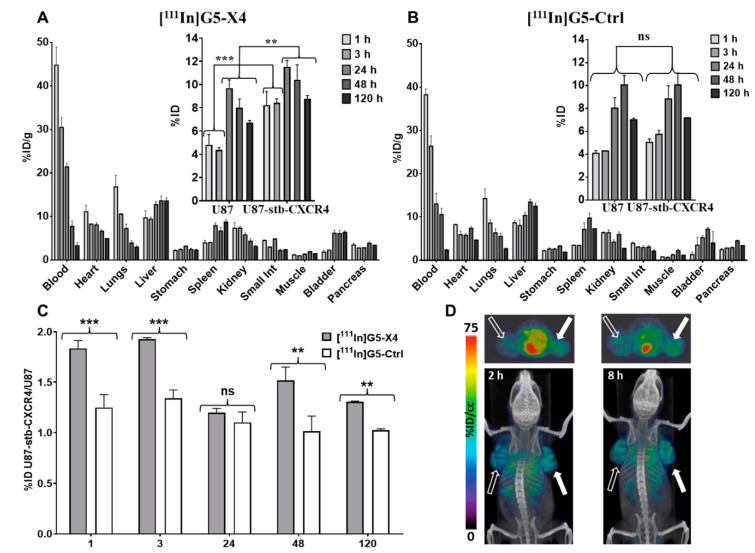
Evaluation of [^111^In]G5-X4 CXCR4 specificity. (**A**,**B**) Ex vivo biodistribution of [^111^In]G5-X4 and [^111^In]G5-Ctrl in NOD/SCID mice bearing U87 and U87-stb-CXCR4 tumors at 1 h, 3 h, 24 h, 48 h, and 120 h after injection (presented as percent of injected dose per gram tissue %ID/g, n = 3 or 4, *** *p* < 0.01, ** *p* < 0.05, ns—non-significant). (**C**) U87-stb-CXCR4/U87 ratios demonstrating faster uptake and enhanced retention of [^111^In]G5-X4 in U87-stb-CXCR4 tumors vs. U87 tumors and [^111^In]G5-Ctrl both tumors. (**D**) Transaxial and volume-rendered SPECT/CT images of mouse bearing U87-stb-CXCR4 (solid arrow) and U87 (hollow arrow) tumors collected 2 and 8 h after injection of [^111^In]G5X4. Imaging results are consistent with ex vivo biodistribution data indicating higher uptake of [^111^In]G5-X4 in U87-stb-CXCR4 tumor compared to U87 tumor with relatively high background at early time points.

**Figure 4 pharmaceutics-14-00655-f004:**
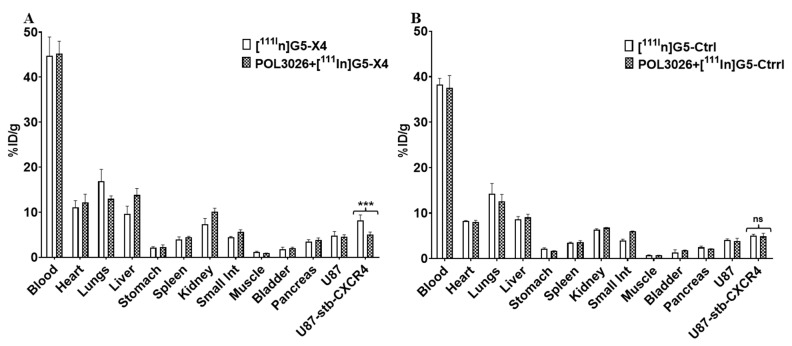
Evaluation of [^111^In]G5-X4 in vivo specificity. For blocking experiments, mice were injected with 45 μg of unmodified POL3036 peptidomimetic 1 h prior to the administration of (**A**) [^111^In]G5-X4 and (**B**) [^111^In]G5-Ctrl radiotracers. Biodistribution was carried out at 3 h after radiotracer injection. Results indicate that uptake of [^111^In]G5-X4 in CXCR4 positive tumors was blocked by POL3036 to the level detected in CXCR4 negative tumors and by [^111^In]G5-Ctrl in both tumors. *** *p* < 0.01.

**Table 1 pharmaceutics-14-00655-t001:** Physicochemical characterization of dendrimer conjugates.

DC	# of NH_2_	# of DOTA	# of PEG_2000_	# of PEG_2_	# of SAPOL3026	MW [*m*/*z*]	Size [nm]	ZP [mV]
1	128	0	0	0	0	26,111	5.57 ± 0.84	45.52 ± 0.74
2	124	4	0	0	0	27,661	4.77 ± 0.44	38.15 ± 024
3	116	4	8	0	0	44,027	8.99 ± 0.76	20.10 ± 0.89
4	54	4	8	62	0	52,250	8.78 ± 0.37	10.04 ± 0.65
G5-Ctrl	54	4	8	62	0	50,666	7.81 ± 0.85	12.41 ± 1.4
G5-X4	54	4	8	62	3	57,250	9.11 ± 0.92	14.9 ± 0.51

DC—dendrimer conjugate; MW—molecular weight derived from the highest signal intensity detected in MALDI-TOF spectra for the main generation dendrimer conjugates; ZP—zeta potential.

## Data Availability

Supporting data to the research can be found in the supplementary materials of this manuscript.

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
