# Peer review of "An Evaluation of CXCR4 Targeting with PAMAM Dendrimer Conjugates for Oncologic Applications"

_pharmaceutics, 2022, doi:10.3390/pharmaceutics14030655_

Round 1

Reviewer 1 Report

Pharmaceutics –(MDPI)- Revised on (2/7/22)

This is important and timely examination of chemokine receptor 4 (CXCR4) as a promising diagnostic and therapeutic target for the management of cancer using dendrimer-based nanotechnology. This work should be of high interest to the Pharmaceutics readership as well as those interested in receptor mediated, active targeting of certain cancer types. This manuscript is reasonably well written, fairly well referenced and appropriately illustrated with suitable graphics. Furthermore, the experimental information provided is adequate to make all conclusions compelling to this reviewer. As such, I recommend publication after first considering several important suggestions which include; the clarification of several confusing communication issues as well as a number of  minor typographical/spelling errors , all of  which if remediated should enhance this manuscript. These items are specifically as described below;

  1. The current title is somewhat indirect and lacks the proper clarity that this very nice work deserves. Perhaps the following title revision would remediate this issue: “An Evaluation of CXCR4 Targeting with PAMAM Dendrimer Conjugates for Oncologic Applications”.
  2. Page 1, line 39; omit of
  3. Page 2, line 49; omit at once
  4. Page 2, line 52; … Insert theses references after straightforward modification:
    Kim et al., Recent progress in dendrimer-based nanomedicine development. Arch. Pharm. Res. 41, 571–582 (2018);
    H. Yang et al., Targeted nanosystems: advances in targeted dendrimers for cancer therapy. Nanomedicine (2016),12:309–316;
    D.A. Tomalia, et.al., Biochemical Society Transactions, 35(1), 61-67, (2007);
    R.M. Kannan, et.al, J. Intern. Med., 276, 579-617 (2014).
  5. Page 4. Line 171-172; This sentence is confusing!! Please clarify!
  6. Page 4, line 157; The U87 (human?? glioblastoma) and…
  7. Page 5, line 199; How many animals were used in this study?? Please clarify!!
  8. Page 6, line 263; …of prepared dendrimer
  9. Table 1; Please define what NP means!!
  10. Page 4, line 152-154; The particle size data obtained for naked G5; PAMAM dendrimers are clearly appropriate. However, it is somewhat confusing to understand the size variations with other formulations discussed in the manuscript using only DLS. As such, could the authors provide reasons for using the number (%) pattern for determination of dendrimer diameters? Why did the authors not try to check or correlate these data with intensity (%) or volume (%)? Could the authors provide either references or appropriate arguments to support the preferred use of number (%) for DLS measurement of dendrimer dimensions?

Author Response

Reviewer 1

This is important and timely examination of chemokine receptor 4 (CXCR4) as a promising diagnostic and therapeutic target for the management of cancer using dendrimer-based nanotechnology. This work should be of high interest to the Pharmaceutics readership as well as those interested in receptor mediated, active targeting of certain cancer types. This manuscript is reasonably well written, fairly well referenced and appropriately illustrated with suitable graphics. Furthermore, the experimental information provided is adequate to make all conclusions compelling to this reviewer. As such, I recommend publication after first considering several important suggestions which include; the clarification of several confusing communication issues as well as a number of  minor typographical/spelling errors , all of  which if remediated should enhance this manuscript. These items are specifically as described below;

  1. The current title is somewhat indirect and lacks the proper clarity that this very nice work deserves. Perhaps the following title revision would remediate this issue:“An Evaluation of CXCR4 Targeting with PAMAM Dendrimer Conjugates for Oncologic Applications”.

The title was changed according to the reviewer’s suggestion

  1. Page 1, line 39; omit of

“of” was deleted

  1. Page 2, line 49; omit at once

“at once” was deleted

  1. Page 2, line 52; … Insert theses referencesafter straightforward modification:
    Kim et al., Recent progress in dendrimer-based nanomedicine development.  Pharm. Res. 41, 571–582 (2018);
    H. Yang et al., Targeted nanosystems: advances in targeted dendrimers for cancer therapy. Nanomedicine (2016),12:309–316;
    D.A. Tomalia, et.al., Biochemical Society Transactions, 35(1), 61-67, (2007);
    R.M. Kannan, et.al, J. Intern. Med., 276, 579-617 (2014).

All above references were added with appropriated modification of the text.

  1. Page 4. Line 171-172; This sentence is confusing!! Please clarify!

The sentence was corrected:

H69 small cell lung cancer (SCLC) cells were detached from growing flask using a non-enzymatic cocktail (Gibco) when confluency reached 50-70%. Detached cells were washed twice with 5 mL of 1xPBS buffer containing 2 mmol EDTA and 0.5% FBS, that was further used for flow cytometry.

  1. Page 4, line 157; The U87 (human??glioblastoma) and…

Yes, U87 human glioblastoma. The entire paragraph was rephrased. 

All cell culture reagents were obtained from Invitrogen (Halethorpe, MD) unless otherwise indicated. The CHO (Chinese hamster ovary), U87 (human glioblastoma), H1155 (non-small cell lung cancer (NSCLC) and H69 (human small cell lung cancer SCLC) cell lines were purchased from American Type Culture Collection (Manassas, VA). A human U87 cell line stably transfected with human  CXCR4 (U87-stb-CXCR4) and CD4 was purchased from the NIH AIDS Research Reference Reagent Program [30].  All cell lines were cultured per manufacturer  protocols. The CHO cell line with stable expression of SNAP-CXCR4 (CHO-SNAP-CXCR4) was generated in our laboratory and maintained as described earlier [31].

  1. Page 5, line 199; How many animals were used in this study?? Please clarify!!

Numbers of animals were added

  1. Page 6, line 263; …of prepared dendrimer
  2. Table 1; Please define what NP means!!

Description of the table was changed: Physicochemical characterization of dendrimer conjugates  NP was substituted with DC and used abbreviations were defined.

DC - dendrimer conjugate, MW - molecular weight derived from the highest signal intensity detected in MALDI-TOF spectra for the main generation dendrimer conjugates, ZP - zeta potential.

  1. Page 4, line 152-154; The particle size data obtained for naked G5; PAMAM dendrimers are clearly appropriate. However, it is somewhat confusing to understand the size variations with other formulations discussed in the manuscript using only DLS. As such, could the authors provide reasons for using the number (%)pattern for determination of dendrimer diameters? Why did the authors not try to check or correlate these data with intensity (%) or volume (%)? Could the authors provide either references or appropriate arguments to support the preferred use of number (%) for DLS measurement of dendrimer dimensions?

Number weighted size distribution represents actual size of nanoparticles present in the solution and their content. This usually correlates well with size of nanoparticles seen under TEM. Very low amounts of larger nanoparticles that strongly scatter laser light may provide misleading results derived using intensity or volume weighted size distribution not reflecting actual composition of the samples.

We also added this statement to the manuscript.

Results in the Table 1 are a mean of three sequential measurements of number weighted size distribution as it most accurately represents size of nanoparticles present in the solution and their content.

Reviewer 2 Report

The article refers to the preparation of G5 modified PAMAM dendrimers with the chemokine receptor CXCR4 and the study of its pharmacokinetics.   

Although the topic is of interest, the manuscript lacks important aspects regarding the characterization of the synthetized compounds, and for this reason, mayor revisions are needed.

Regarding the synthesis and characterization of the manuscript:

- Figure 1 and Figure 2 are swapped.

- In Figure 2, “pipiridine” should be corrected.

- All modifications made over PAMAM G5 dendrimers have been only characterized in terms of DLS, Z-potential and MALDI.

Size distribution of all compounds should be shown in the ESI (and not only the corresponding to G5-X4). Moreover, more experiments to ensure the mono-dispersity of the prepared compounds are needed. How do the authors confirm that only one specie has been obtained? In MALDI spectra is shown the peak showed in table 1 of the manuscript, but is it possible the presence of other species?

In other words, How do the authors demonstrate that only 4 units of DOTA are inserted in compound 2? Could it be a mixture G5-(DOTA)1, G5-(DOTA)2, G5-(DOTA)3, etc…..?

Author should demonstrate with more experiments that the prepared compounds are mono-disperse. The compounds should be characterized by HPLC, NMR, etc…also Diffusion NMR experiments could help.

Also, complete MALDI spectra should be included in the Supporting Information

Author Response

Reviewer 2 

The article refers to the preparation of G5 modified PAMAM dendrimers with the chemokine receptor CXCR4 and the study of its pharmacokinetics.   

Although the topic is of interest, the manuscript lacks important aspects regarding the characterization of the synthetized compounds, and for this reason, mayor revisions are needed.

We like to thank the reviewer for his/her insightful comments. In fact, we agree that thorough characterization of synthetized dendrimer conjugates would provide useful information but this is beyond the scope of the paper. Our study objective was to formulate CXCR4-targeted dendrimers and their detailed in vitro and in vivo characterization. We measured average number of conjugated functionalities in synthetized conjugates and their size distribution that provides the most useful information to understand their pharmacokinetics.

Regarding the synthesis and characterization of the manuscript:

- Figure 1 and Figure 2 are swapped.

We thank reviewer for pointing it out. Although all figures were correctly prepared in the originally submitted manuscript

- In Figure 2, “pipiridine” should be corrected.

That was corrected. The font denoting average number of conjugated functionalities upon each modification was also increased for better visibility.

- All modifications made over PAMAM G5 dendrimers have been only characterized in terms of DLS, Z-potential and MALDI.

Size distribution of all compounds should be shown in the ESI (and not only the corresponding to G5-X4). Moreover, more experiments to ensure the mono-dispersity of the prepared compounds are needed. How do the authors confirm that only one specie has been obtained? In MALDI spectra is shown the peak showed in table 1 of the manuscript, but is it possible the presence of other species?

Size distributions were added into Supplementary Information

Figure s2. Representative number weighted size distribution acquired for starting dendrimer and all synthetized conjugates clearly demonstrating increase of the size of dendrimer after PEGylation

We did not claim that PAMAM dendrimers and formulated conjugates are monodispersed in the manuscript. Actually, PAMAM dendrimers show low polydispersity. Commercially available technical grade of PAMAM dendrimers contain 3 different species trailing generation (3-5 %, generation 5 contains generation 4), main generation (85-90%) and dimers (5-10 %). As presented in the literature [1] PAMAM dendrimers possess two other imperfections skeletal (missing arms and intramolecular loops) and substitutional in addition to generational imperfection leading to the presence of multiple different species in the PAMAM samples.  For this reason, peaks in MADLI-TOF spectra are broad and as presented in the supporting information get broader upon each modification step. We focused our attention on main generation and used highest intensity signal of the peaks to calculate average number of conjugated moieties.   

This MALDI-TOF spectrum of PAMAM generation 5 dendrimer (included in the supporting information) for starting material shows presence of trailing generation, main generation and dimers. However, MALDI-TOF is not quantitative and cannot be used to calculate content of species in the sample. Estimation of the polydispersicy would require application of specialized equipment such as size exclusion chromatography connected with multi-angle light scattering and differential refractive index detectors to obtained molecular weight distribution, Mw and Mn to derive polydispersity index (Mw/Mn) [2] which is beyond the scope of this paper.  

[1] Shi, X.Y., et al., Generational, skeletal and substitutional diversities in generation one poly (amidoamine) dendrimers. Polymer, 2005. 46(9): p. 3022-3034.

 [2]Lesniak, W.G., et al., Synthesis and characterization of PAMAM dendrimer-based multifunctional nanodevices for targeting alpha(v)beta(3) integrins. Bioconjugate Chemistry, 2007. 18(4): p. 1148-1154.

In other words, How do the authors demonstrate that only 4 units of DOTA are inserted in compound 2? Could it be a mixture G5-(DOTA)1, G5-(DOTA)2, G5-(DOTA)3, etc…..?

We covered that above

To address reviewer’s critiques we modified description of results accordingly:

Commercially available of PAMAM dendrimers contain three different species trailing generation, main generation (usually approximately 90%) and dimers [35, 36]. Presence of all three species was confirmed in the spectrum of the dendrimer that we used as starting material (Supporting Information Figure s1) with the most pronounced peak for main generation-5 dendrimer.  The change in molecular weight observed for main generation-5 dendrimer upon each synthetic step was used to calculate the average number of covalently attached functionalities.

Author should demonstrate with more experiments that the prepared compounds are mono-disperse. The compounds should be characterized by HPLC, NMR, etc…also Diffusion NMR experiments could help.

Such extensive characterization of formulated conjugates is beyond the scope of this paper

We recorded 1H NMR spectra for the starting dendrimer and formulated conjugated, however due to extensive signals overlapping they are not informative, as shown below we decided not to include them in the manuscript.

Also, complete MALDI spectra should be included in the Supporting Information

Covered above

Reviewer 3 Report

The authors an interesting study concerning  a chemokine receptor 4 (CXCR4) as an active molecule used for targeted formulations for cancer therapy. There are some remarks that should be revised: 
(1) line 46-49 - decrease the font
(2) why in cells lined description is not included H69 cell line? The authors finally used three cell lines, and there is no justification as, why some experiments are made on glioma and CHO cells and the rest on lung cancer cells. Additionally, H1155 cells were used in vivo. It should be clearly presented. 
(3) IC50 in the abstract is for which cell line?
(4) the abstract is too general; as two cancer types were used and one normal cell line, it should be highlighted the antitumor effect. 
(5) references - change font

Author Response

Reviewer 3

The authors an interesting study concerning a chemokine receptor 4 (CXCR4) as an active molecule used for targeted formulations for cancer therapy. There are some remarks that should be revised: 

(1) line 46-49 - decrease the font

The font in the originally submitted manuscript (word document) was corrected.

(2) why in cells lined description is not included H69 cell line? The authors finally used three cell lines, and there is no justification as, why some experiments are made on glioma and CHO cells and the rest on lung cancer cells. Additionally, H1155 cells were used in vivo. It should be clearly presented. 

Description of H69 human small cell lung cancer was added.

To evaluate affinity of our conjugates to CXCR4 we used proprietary assay from Cisbio Bioassays that requires use of CHO-K1-SNAP-CXCR4 cells as described in the referenced paper. We used human glioblastoma U87 and isogenic U87-stb-CXCR4 cell lines as they can inoculated in the same mouse and provide xenografts with low to no, and high CXCR4 expression, respectively for appropriate evaluation of in vivo target specificity. For evaluation of chemotaxis inhibition, we initially used U87-stb-CXCR4 but these are too large for this essay and we used smaller H69 cells.  For this reason, we included these results in Supplementary information. 

Prompted by the relatively high uptake of non-targeted control dendrimers in U87 and U87-stb-CXCR4 tumors due to passive targeting, we also intended to evaluate them using clinically relevant orthotopic model of H1155 human non-small cell lung cancer.

(3) IC50 in the abstract is for which cell line?

CHO-SNAP-CXCR4 cells

Abstract was modified accordingly:

G5-X4 demonstrated an IC50 of 0.95 nM to CXCR4 against CXCL12-Red in CHO-SNAP-CXCR4 cells

(4) the abstract is too general; as two cancer types were used and one normal cell line, it should be highlighted the antitumor effect. 

More information was included in the abstract. However, considering a 200 word limit, we could not include all details in the abstract

(5) references - change font

Font was changed

Round 2

Reviewer 2 Report

The article refers to the preparation of G5 modified PAMAM dendrimers with the chemokine receptor CXCR4 and the study of its pharmacokinetics.

Although the topic is of interest, the manuscript lacks important aspects regarding the characterization of the synthetized compounds, and for this reason, mayor revisions are needed.

We like to thank the reviewer for his/her insightful comments. In fact, we agree that thorough characterization of synthetized dendrimer conjugates would provide useful information but this is beyond the scope of the paper. Our study objective was to formulate CXCR4-targeted dendrimers and their detailed in vitro and in vivo characterization. We measured average number of conjugated functionalities in synthetized conjugates and their size distribution that provides the most useful information to understand their pharmacokinetics.

I completely disagree at this point. If I understand correctly, the scope of the manuscript is the development of a new formulation for oncologic applications. No agency will allow the in vivo application of a system that is not correctly characterised. In this sense, in my opinion not only characterization is needed, but also the reproducibility in the synthetic procedure should be demonstrated.

The described compounds should be more characterized or at least, author should not claim the exact composition of each compound. With the data provided, authors cannot claim that they exactly introduce 4 units of DOTA, 8 units of PEG-Fmoc, 62 units of PEG, etc….

Author Response

In the response to the reviewer comments we modified Figure s1 included in Supporting Information, added reverse-phase high performance liquid chromatography (Supporting Information,  Figure s2) and 1H NMR (Supporting Information,  Figure s3), and modified Results section accordingly:

Starting dendrimer, and synthetized conjugates were characterized by MALDI-TOF, RP-HPLC, 1HNMR, DLS and zeta potential (Table 1 and Supporting Information Figure s1, s2, s3 and s4). Commercially available of PAMAM dendrimers contain three different species trailing generation, main generation and dimers [34, 35]. All three species were detected in the spectrum of the dendrimers that we used as starting material (Supporting Information Figure s1) with the most pronounced peak for main generation-5 dendrimer at 26111 Da. Trailing generation and dimers were detected at 12600 and 50950 Da, respectively. Increase of the molecular weights indicated conjugation of on average two, four and five DOTA molecules with trailing generation, main generation and dimers, respectively. Starting from conjugates III trailing generation, main generation-5 and dimers could not be resolved, showing broad peak indicating wide distribution of conjugates present in the sample. Shift of the highest signal intensity detected for this peak was used to calculate the average number of covalently attached PEG2000, mPEG2 and SA-POL3026 moieties. In agreement with MALDI-TOF RP-HPLC showed presence of trailing generation, main generation and dimers with relative percent of 3.5, 90.9 and 5.6%, respectively in dendrimers used as starting material (Supporting Information Figure s2). RP-HPLC of G5-Ctrl and G5-X4 showed significant broadening of the peaks and shift to longer retention with no of trailing generation, main generation and dimers confirming broad distribution of the conjugates. 1H NMR further confirmed modifications of dendrimers (Supporting Information Figure s3). DLS analysis indicated an increase in dendrimer size upon PEGylation by approximately 3 nm and a significant reduction in net surface positive charge (Supporting Information Figure s4). G5-Ctrl and G5-X4 dendrimers had size distribution with 7.81 ± 0.85 nm and zeta potential of 12.41 ± 1.43 mV and 9.11 ± 0.92 nm and zeta potential of 14.93 ± 0.51 mV, respectively.        

The characterization of dendrimer conjugates provided in this manuscript is in line with all other dendrimer work carried out in this field. These are large globular macromolecules (nanoparticles) that are routinely characterized by MALDI-TOF mass spectrometry, and HPLC. In depth NMR spectroscopy of nanoparticles with large number of protons is primarily used to study the overall effects of conjugated moieties on proton shifts. NMR is simply not a technique to use to calculate the number of small, low molecular weight ligands in a much larger nanoparticle with numerous hydrogens. Deciphering specific coupling interactions out of tens of thousands of possible overlapping interactions is simply not feasible and falls far outside the scope of this manuscript as well as the field of oncology. That is why mass spectrometry is the primary instrumentation for this application. In case of small organic scaffolds (i.e. low molecular weight compounds), we completely agree that thorough NMR spectroscopy is required. However, this is not the case with what we are reporting in this study as we are using commercially available starting materials that have been well studied and reported, as well as routine conjugation chemistry which has been long established in literature. We have provided complete characterization data that is widely accepted in this field. We do, however, completely agree with the reviewer that the number of attached ligands is an average number for a specific generation (e.g. main generation) and not an exact number. Therefore, in addition to providing more details, we have also clarified the language in the manuscript to better represent this notion. We hope that these additional modifications help clarify this concept.

Round 3

Reviewer 2 Report

The manuscript can now be accepted for publication